# Multi-Modal Long-Term Person Re-Identification Using Physical Soft Bio-Metrics and Body Figure

**Nadeen Shoukry** [1] , **Mohamed A. Abd El Ghany** [2,3] **and Mohammed A.-M. Salem** [4,5,*]

1   Media Engineering and Technology Department, The German University in Cairo, Cairo 11835, Egypt; nadeen.shoukry@gmail.com
2   Information Engineering and Technology Department, The German University in Cairo, Cairo 11835, Egypt; mohamed.abdel-ghany@guc.edu.eg or mohamed.salem@ies.tu-darmstadt.de
3   Integrated Electronic Systems Lab, TU Darmstadt, 64283 Darmstadt, Germany
4   Digital Media Engineering and Technology Department, The German University in Cairo, Cairo 11835, Egypt
5   Scientific Computing Department, Faculty of Computer and Information Sciences, Ain Shams University, Cairo 11835, Egypt
*   Correspondence: mohammed.salem@guc.edu.eg

**Abstract:** Person re-identification is the task of recognizing a subject across different non-overlapping cameras across different views and times. Most state-of-the-art datasets and proposed solutions tend to address the problem of short-term re-identification. Those models can re-identify a person as long as they are wearing the same clothes. The work presented in this paper addresses the task of long-term re-identification. Therefore, the proposed model is trained on a dataset that incorporates clothes variation. This paper proposes a multi-modal person re-identification model. The first modality includes soft bio-metrics: hair, face, neck, shoulders, and part of the chest. The second modality is the remaining body figure that mainly focuses on clothes. The proposed model is composed of two separate neural networks, one for each modality. For the first modality, a two-stream Siamese network with pre-trained FaceNet as a feature extractor for the first modality is utilized. Part-based Convolutional Baseline classifier with a feature extractor network OSNet for the second modality. Experiments confirm that the proposed model can outperform several state-of-the-art models achieving 81.4 % accuracy on Rank-1, 82.3% accuracy on Rank-5, 83.1% accuracy on Rank-10, and 83.7% accuracy on Rank-20.

**Keywords:** FaceNet; long-term person re-identification; OSNet; PCB; PRCC dataset; Siamese network

## 1. Introduction

City streets are not only numbers and statistics; they are homes to people and families who should feel safe walking down the streets at any time of the day. Over fifty years ago, writer and journalist Jane Jacobs devoted her efforts to studying the concept of "eyes on the street". People feel safe and secure in public spaces, despite being among strangers whom she refers to as the eyes on the street [1].

Our work intends to make all public streets safe, not only crowded ones for every single person. Person re-identification is the process of associating images or videos of the same person across different cameras at different times [2] ; it is a non-invasive biometric surveillance tool, meaning the process of re-identifying a person does not rely on identity but soft biometrics such as height, gender, age. The applications of person re-identification systems are indispensable such as visual impairment assistance systems, forensic search, and long-term multi-camera tracking. Person re-identification has not received much attention from the scientific community recently due to the challenges facing it, such as the scarcity of publicly available large datasets, most of the datasets are captured with CCTV cameras producing images with low resolution. Other challenges are related to the camera positioning: illumination changes, partial or total occlusion of a person and pose variation. This problem remains unsolved and challenging for several reasons. Illumination changes:

indoor lighting reflected light and shade from colored surfaces can cause the same person to appear differently across multiple cameras. Low resolution; old CCTV cameras produce images with low-resolution. Occlusion: in crowded environments presents a challenge in extracting the subjects' features. Lastly, uniform clothing: implementing a re-identification system in a school or a factory where people are wearing the same clothes can make it harder to extract discriminative features [3–7]. To build a model that generalizes well on the previously mentioned challenges, the model must be trained on a dataset that includes images addressing those challenges. The main objective of our work is to automate the task of person re-identification making all public streets safe.

The main contributions of our work are:

- A novel approach is presented to address the problem of person re-identification under cloth-changing conditions by combining different modalities to achieve the task. The first modality is the facial region and the second modality is the remaining body part. A separate neural network was built for each modality to extract appropriate features, the resulting features from both modalities were combined to produce an informative decision using score-level fusion module.
- The proposed methodology was tested on a challenging dataset, PRCC dataset is one of the few public datasets that incorporate cloth-changing.
- The approach is compared against recent state-of-the-art models. Also to our knowledge, this is the first result of the MLFN model on the PRCC dataset.

The work is organized as follows: in Section 2, an overview of the related work is presented. Section 3, describes the dataset used, pre-processing applied to images and the proposed model. Section 4 presents the experiments conducted along with their results. Conclusions and future work are presented in Sections 5 and 6, respectively.

## 2. Related Work

This section is an overview of the previous work done in this field. Before the advancement of deep learning, the scientific community addressed the problem using traditional methods consisting of two components: extracting features from images and a distance metric to compare those features. Deep learning approaches are evolving and showing outstanding performances in numerous applications. Therefore, it is heavily utilized in the person re-identification task. Deep learning approaches to tackle this problem are divided into three main categories: CNN-based models, multi-scale deep learning models, and part-based deep learning models.

The approaches presented in this section are the main inspiration to our work.

### 2.1. CNN-Based Multi-Modal Deep Learning for Person Re-Identification

Ja Hyung et al. [8] propose a multi-modal human recognition method that uses both face and body to recognize a person. The proposed model is a fusion of two networks VGG Face-16 and ResNet-50. VGG Face-16 is responsible for processing the facial regions in the image. ResNet-50 is used for processing the body regions in the image. VGG Face-16 is a suitable fit for the task since it is a deep CNN model that was trained on a database for face recognition task [9]. They used ResNet-50 since the body region has more texture, color, and shape. The final decision of the model is formed using score-level fusion and the weighted sum rule is applied to improve the recognition performance.

The harmonious attention CNN (HACNN) was proposed [10] in 2018. In their work, they present a novel idea of learning attention selection features jointly with feature representation. In an attempt to optimize the person re-identification task using a lightweight architecture with fewer parameters.

Multi-level factorisation Net (MLFN) presented by [11] in 2018. The novel learning process presented in their work depends on learning identity-discriminative and view-invariant visual factors at multiple semantic levels.

Spatial and channel partition representation network (SCR) proposed by [12] in 2020. Their proposed framework exploits both spatial and channel information. Their pipeline

starts with a strong backbone network ResNet-50. They applied some modifications to ResNet-50 to extract more that e high-level features can be kept in the feature map. The modifications are as follows: (a) replacing the down-sampling done using stride-2 convolution by a stride-1 convolution in the conv5 1 layer, (b) all the layers after conv4, 1 layer are duplicated to form 3 independent branches. The backbone feature extractor network is followed by Multiple Spatial-channel Partitions in a Pyramidal structure which is applied to utilize all the information in the feature map. A pyramid structure is used to train global, spatial and channel partitioned features separately. Experiments to evaluate SCRNet were conducted on three state-of-the-art datasets Market-1501, DukeMTMC-Reid, and CUHK03, and one video-based dataset. SCRNet outperforms the previous state-of-the-art in both single and cross-domain re-identification tasks.

ResNet50 is an advancement of the original ResNet model (short for Residual Networks) has 48 convolutional layers along with 1 MaxPool and 1 Average Pool layer. ResNets are used as a backbone for several computer vision tasks and are known to achieve remarkable results [13]. Deep networks are hard to train because of the notorious vanishing gradient problem, as the gradient is back-propagated to earlier layers, repeated multiplication may make the gradient extremely small. As a result, as the network goes deeper, its performance gets saturated or starts degrading rapidly. ResNets provide a framework that made it possible to train extremely deep neural networks (150+ layers) that can contain hundreds or thousands of layers and still achieve exceptional performance. The contribution ResNets made to address the vanishing/exploding gradient problem by introducing skip connections allowing the gradient to flow through an alternate shortcut path. Skip connections allow the model to learn an identity function which ensures that the higher layer will perform at least as good as the lower layer, and not worse [14].

### 2.2. Multi-Scale Deep Learning Architectures for Person Re-Identification

Xuelin, Qian et al. [3] proposed an advanced multi-scale deep learning model (MuDeep) that can re-identify a person by extracting discriminative features at multiple scales with automatically determined scale weighting. The idea behind multi-scaling is to mimic how a human brain re-identifies people from coarse to fine. For example, some people are distinguishable by their global features (e.g., gender, body, etc.). On the other hand, some people are distinguishable by their local features (e.g., shoes, bags, etc.). They based their model on the fact that people are distinguishable at correct spatial locations and scales. One might identify a person from an obvious detail like glasses. Sometimes a closer look at the fine details (e.g., eye colour) is needed to identify a person. The architecture of the network is a Siamese one [15]. The main contributions presented in this work are:

- Tied Convolution for pre-processing the input
- Multi-scale stream layers: Analyses the data stream with several receptive field sizes.
- Saliency-based learning fusion layer: To fuse the output streams from the previous layers.
- Sub-nets for person re-identification: Used for verification so the problem can be transformed into a binary classification one, stating whether a pair of images represent the same person or not.

Zhou et al. [16] proposed Omni-Scale Feature Learning for Person Re-Identification. They utilized Omni-scale features in the person re-identification task. The idea behind Omni-scale feature learning is combining features of multiple scales, OSNet is a network that specializes in that domain. Unlike Xuelin Qian et al. [3] Omni-scale features include heterogeneous and homogeneous features as well as features at multiple scales. In other words, homogeneous features are global features (young man, white shirt, grey shorts, etc.) and local features (trainers or sandals). Heterogeneous features take a closer look at the complicated and richer details like a specific logo on the white shirt. The key to Omni-scale feature is to combine both homogeneous and heterogeneous features. A white shirt on its own or a specific logo is not discriminative or enough to identify a person.

OSNet starts by decomposing the convolutional layer and then introduces a full-size residual block and a unified aggregation gate. The basic building block of the OSNet is made of multiple convolutional streams each with a different receptive field (different scales). The basic building block of OSNet is the lite $3 \times 3$ convolutional layer, which is given by Equation (1). The block receives input $x$ and learns $x'$.

$$x' = F(x) \tag{1}$$

Inspired by the Inception model architecture [17] to achieve multi-scale feature learning, several lite $3 \times 3$ convolutional blocks are stacked together to achieve a lightweight network. As shown in the Equation (2) to stack several $3 \times 3$ blocks function $F$ is extended by introducing a new dimension, exponent $t$, which represents the scale of the feature. For $t > 1$, stacking $t$ Lite $3 \times 3$ layers, results in a receptive field of size $(2t + 1) \times (2t + 1)$.

$$x' = \sum_{t=1}^{T} F^t(x), \, s.t. \, T \geq 1 \tag{2}$$

The resulting multi-scale feature maps from the previously mentioned convolutional stream are dynamically fused through unified aggregation gates. The aggregation gates are subnets that share parameters across all streams to generate weights for each channel in the convolutional stream. Training those parameters improves the weight given to each channel. In other words, if during the training process back-propagation of the error shows that the stream resulting from a receptive field of three contributes with more valuable information to the model's performance, the aggregation gates will assign a dominant weight to this stream and less weight to the other streams. The architecture of OSNet is relatively small and lightweight compared to the other networks that are often used in this task, such as ResNet50. Its lightweight nature makes it less prone to over-fitting on small and medium datasets. This property is beneficial since most of the publicly available and state-of-the-art person re-identification datasets are of small or medium size.

### 2.3. Part-Based Deep Learning for Person Re-Identification

Yifan Sun et al. [18] the work proposed in this paper depends on part-level features. Part-level feature extraction is the process of partitioning an image to several parts, followed by extracting features from each part individually and finally combining the learned information from each part. This methodology of feature extraction is known to provide features with great attention to detail. The main contributions offered in this work are the part-based convolutional baseline network and a refined part-pooling method. Given an input image, the main functionality of the Part-based Convolutional Baseline (PCB) network is to output a convolutional descriptor consisting of several part-level features. PCB needs a backbone network to transform the input image into a feature vector. Any network can be used as the backbone to PCB, for example, ResNet50. ResNet50 acts as a backbone to the PCB model after removing the Global Average Pool Layer and subsequent parts. After the image passes through the previously mentioned layers of the ResNet50, it becomes a 3D tensor **T** of activations. Partitioning is uniform (partitions have equal size). Conventional average pooling partitions **T** into $p$ horizontal stripes ($p = 6$, implies that an image is divided into six horizontal strips thus extracting features from 6 different regions). Column vectors of each stripe are averaged into a single part-level column vector **g**: $\mathbf{g}_i (i = 1, 2, ..., p)$.

Dimensionality reduction is applied to reduce the dimensions of **g** to 256 the new column vector is called **h**: $\mathbf{h}_i (i = 1, 2, ..., p)$. Finally each **h** is fed to a classifier implemented using a simple fully connected layer followed by Softmax function to predict the identity of the input. Uniform partitioning is simple and effective. However, it has some drawbacks; it introduces inevitable outliers. Relocating outliers in their correct partitions will overcome this problem. Therefore a refined part-pooling module (RPP) was introduced, simply it

measures the similarity between a given part $f$ and each part. After that, the RPP module places each part in its correct partition according to the similarity measure.

### 2.4. Face Detection Models

In 2015 researchers at Google proposed a face recognition system called FaceNet FaceNet can outperform several state-of-the-art face recognition methodologies on a wide range of benchmark datasets. The Inception model inspired the architecture of FaceNet. FaceNet is used to extract high-quality facial features, often referred to as face embedding. The network was trained on the Labelled Faces in the Wild (LFW) Dataset [19], a large dataset to learn variance in the pose, viewpoint, and illumination. FaceNet was able to achieve 99.63% on (LFW).

### 2.5. Discussion

Previous work in this field mainly focused on short-term person re-identification. Meaning; a model can re-identify a person using images/videos taken on the same day, but the model's performance will dramatically deteriorate when trying to re-identify the same person after a long period if the persons' clothing changes [20]. Most of the previous work relies heavily on clothing features because models extract features from the entire body image. Long-term re-identification has been understudied by the scientific community due to the limited availability of large-scale public datasets [20].

To improve recognition even after a subject has changed clothes, we experiment with a multi-modal approach inspired by [8]. Instead of feeding the entire input image of a person to one neural network, we consider the upper region of the human body, which includes hair, face, neck, and part of the chest as the first modality, and the remaining body figure as the second modality.

The idea of multi-modal learning stems from how humans learn; when a number of our senses are heightened and engaged during learning, we tend to learn better and retain more information. Building models that learn from different sources of information and then fusing their output improves the overall learning process thus, improving performance.

The upper region of the human body contains information that does not change dramatically over time compared to the body figure changing rapidly with the clothes variation. Therefore , two separate models were built  to learn appropriate features from each part. We feed each modality (face and body) to different neural networks that can extract features from each modality and combine the results of both neural networks. The first modality was extracted from soft bio-metric features.

By definition, soft bio-metrics are the characteristics of a human extracted from images or videos. Soft biometrics are behavioral and physical characteristics used to identify, verify, and describe human subjects. Some examples of soft bio-metrics are height, weight, and facial measurements. To obtain the first modality, after extracting the facial region, we included the hair, face, neck, shoulders, and a portion of the chest of a human subject. Generally, the previously mentioned regions tend not to change as drastically over time as clothing, so they were grouped. After that, the facial region was extended to incorporate part of the hair, neck, shoulders, and a portion of the chest to increase the features extracted from this region. The second modality is the remaining body figure.

## 3. Methodology

### 3.1. Dataset

Some of the datasets that are commonly used for this task are Market-1501 [21], CUHK(01,03) [22], DukeMTMC-reID [23] and VIPER [24]. The previously mentioned datasets do not incorporate cloth variation therefore not suitable for testing the proposed approach. After extensive research in this area, we were able to investigate some of the publicly available datasets such as BIWI [25] and PRCC [26].

BIWI contains 28 different identities captured doing various motion routines using a Microsoft Kinect for Windows. The Kinect provides images with very high resolution,

i.e., 1280 × 960 pixels. The Kinect was capable of providing depth images, segmentation maps, skeletal data, and RGB images. The set-up of collecting this dataset did not match the environment envisioned for deploying our system; the size of the dataset and number of subjects is relatively small. Additionally, the Kinect produces high-resolution images compared to the security cameras often installed in security systems.

In 2020, Qize Yang et al. [26] proposed a new dataset called Person re-identification under moderate Clothing Change (PRCC) dataset, which was used to test our methodology. PRCC contains 221 different identities and a total of 33,698 images which is almost ten times larger than BIWI in terms of the number of identities. The dataset was collected using three cameras. Cameras A and B capture a person wearing the same clothes, camera C captures the same person on a different day with different clothing. Figure 1 contains three images of the same person the first two images on the left show the person wearing the same clothes (pink T-shirt with a colorful logo along with grey pants and backpack), the last image is from camera C where the person is wearing different clothes (black T-shirt with a white logo along with grey pants and a backpack).

The challenging aspects that these dataset meets are not limited to clothes variation only. It takes pose variation, occlusion, viewpoint variation, and illumination changes into consideration.

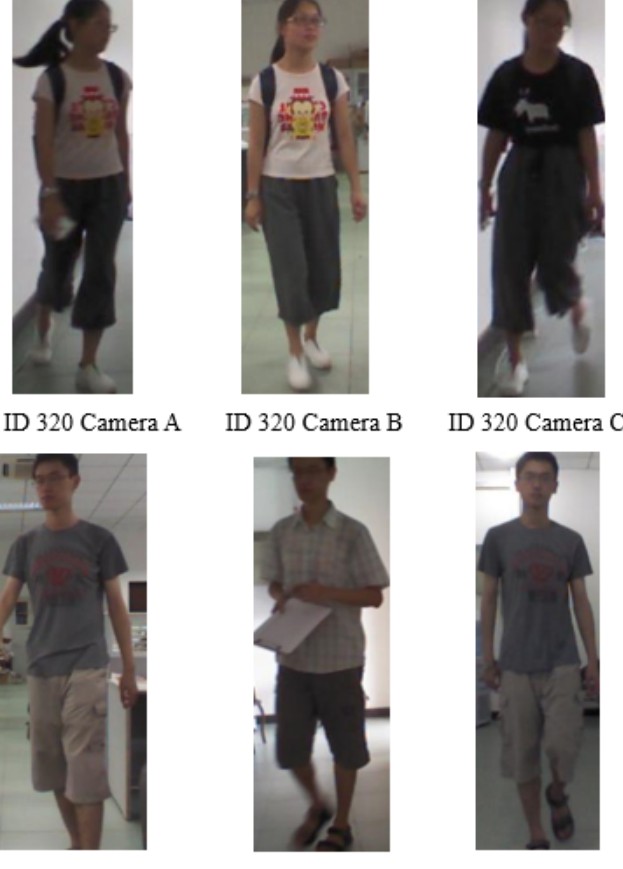

**Figure 1.** Samples of PRCC dataset [26] illustrating clothes variation in images taken of the same subject.

The dataset comprises a total of 221 identities. The experimental setup used 150 identities for training and 71 identities for testing, with no identity overlap between training and testing. Table 1 illustrates how we split the data. For testing, a query set (probe/target images) is needed. The query set evaluates the model's performance according to its ability to match a query image of a person with their pictures from the gallery set (in our setup, query and gallery identities do not overlap). To construct the query set, we sampled two

random images for each identity from each camera, a total of six query images per identity. The size of the query set is 71 × 6 = 426 query images.

**Table 1.** PRCC dataset distribution.

| Subset | IDs | Images | Cameras |
|---|---|---|---|
| Train | 150 | 22,898 | 3 |
| Gallery | 71 | 10,374 | 3 |
| Query | 71 | 426 | 3 |

*3.2. Data Pre-Processing*

Some of the latest models offer some complicated approaches to address this problem, such as [20] in their work, they offer a shape embedding module along with a cloth-eliminating module to disregard the unreliable clothing features and focus only on the body shape information. However, the work presented will follow a lightweight approach to extract features from soft bio-metrics: hair, face, neck, shoulders, and part of the chest. The previously mentioned features are more reliable and robust than changing clothing features. The second modality considers the remainder of the image (body figure).

*3.3. Face Detection*

A face detector (Faceboxes) was used for engineering the facial features. Faceboxes is a CPU real-time face detector with high accuracy. The lightweight nature of the face detector does not affect its performance on accuracy. It can balance the trade-off between accuracy and speed with high performance on both. The structure of the network consists of Rapidly Digested Convolutional Layers (RDCL), Multiple Scale Convolutional Layers (MSCL) [27]. The pre-trained model was used; this model was trained on 12,880 images of the WIDER FACE dataset [28]. One of the limitations of Faceboxes is that it does not detect small faces. Most of the benchmark person re-identification datasets are captured from security cameras, producing images with low resolution and small size. We overcame this problem by resizing the input images before feeding them to the Faceboxes detector.

*3.4. Extending Features*

After detecting faces used to extract facial features, our algorithm was extended to fit more features such as hair, neck, shoulders, and part of the chest. The bounding box was enlarged to include more features resulting from the face detector. The length of the idealized human figure is approximately eight heads tall divided as follows [29]:

- Head;
- From the bottom of the head to the middle of the chest;
- From the middle of the chest to the navel;
- From the navel to the upper edge of the pubis;
- From the upper edge of the pubis to the middle height of the thigh;
- From the middle height of thigh to the middle height of the calf;
- From the middle height of calf to the point below the ankles;
- From the point below the ankles to the feet.

From the second point, we can deduce that the bottom of the bounding box has to be extended with the same length as the face/head to include the neck, shoulders, and part of the human torso. To cover part of the hair, we extended the bounding box from the top to the highest coordinate in the image. Since the images are already produced by human detectors, this extension excludes unnecessary background information. The extension will not only include regular hairstyles but also extreme hairstyles (e.g., high ponytails), hijabs, and turbans. According to [29], the width of the idealized human head is 2/3 times its height. Consequently, the width of the bounding box was extended with 70% of the length of the head to include the shoulders. Pre-processing stages are demonstrated on a sample from the dataset in Figure 2.

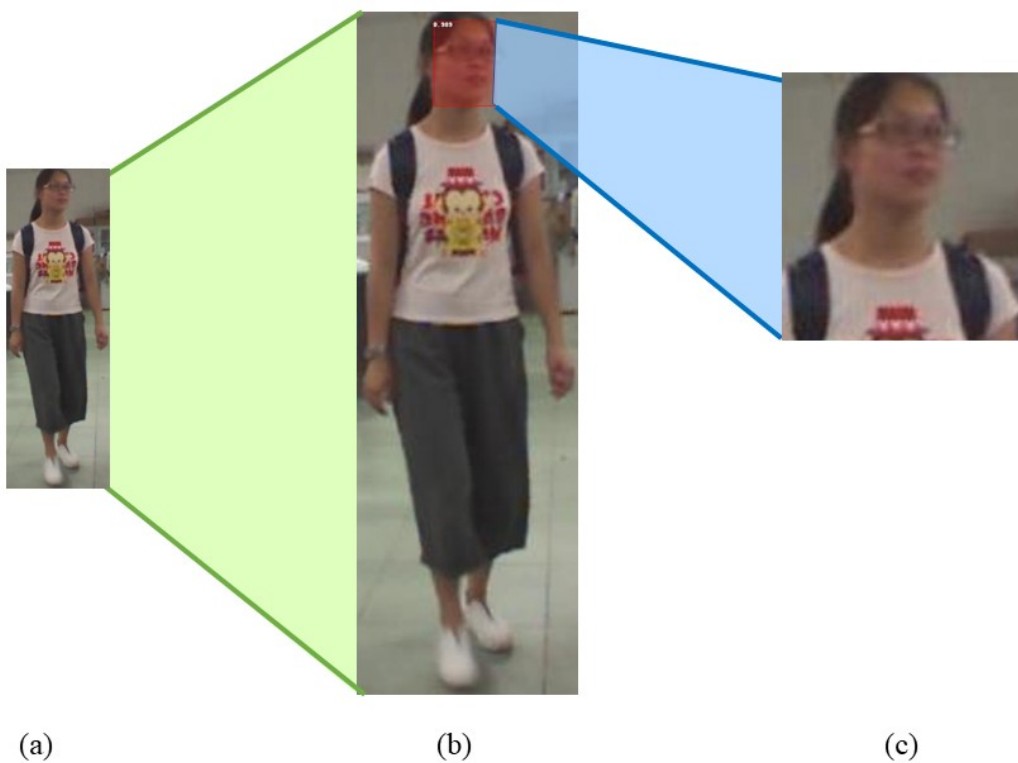

(a)         (b)         (c)

**Figure 2.** (**a**) A sample from PRCC dataset (117 × 361) (**b**) A sample from PRCC dataset after resizing to (351 × 1083) and detecting face using Faceboxes detector (**c**) A sample from PRCC dataset after extending the bounding box around the detected face to include hair, neck, and part of the chest.

### 3.5. Proposed Architecture

Most of the previous state-of-the-art models often used for person re-identification receive the entire body image as input [10,30–32]. A multi-modal human recognition is proposed, rather than feeding the entire image of the body to a CNN. The facial region is extracted using the previously described face detector (Faceboxes) and separated from the body figure. The body figure is the second modality. Figure 3 provides visualization for the entire architecture.

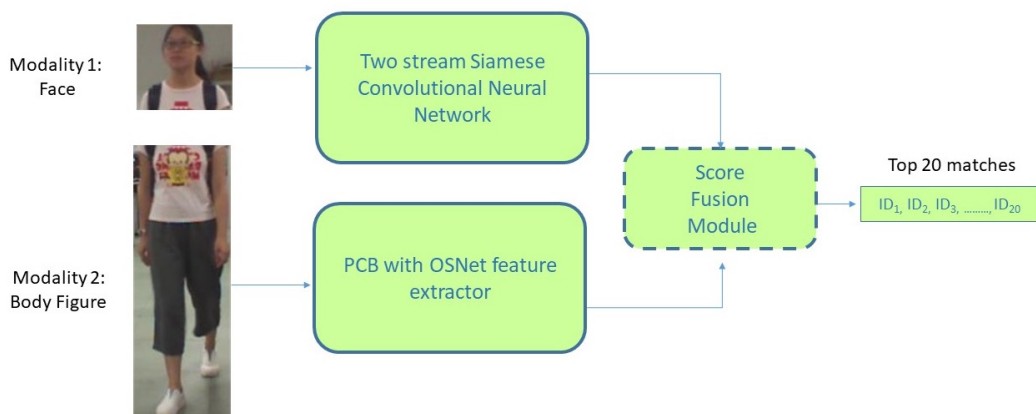

**Figure 3.** Visualization of the proposed architecture.

For the first modality, The proposed two-stream Siamese Network illustrated in Figure 4. Siamese neural networks (sometimes called a twin neural network) are two-stream neural networks used to learn local features and find similarities between pairs of images. Since the two branches of the network are identical (weights are shared), meaning

the network uses the same weights while working on two different input images so that the resulting output feature vectors are comparable. The similarity between feature vectors is often determined using a distance metric such as Euclidean distance, the Mahalanobis distance, Jaccard distance, etc. [33] The proposed model, each branch starts with a pre-trained model for feature extraction to extract high-level features. After that, the distance between the two feature vectors produced is determined using the Euclidean Distance [34].

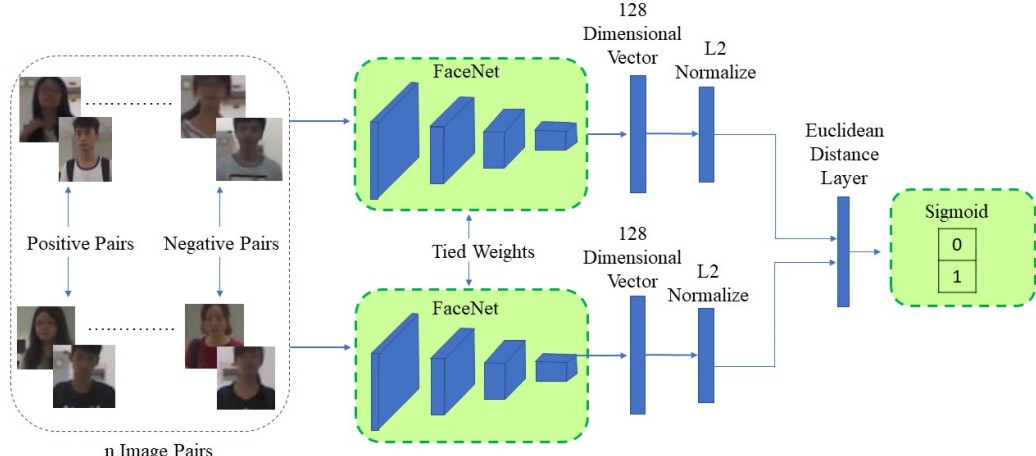

**Figure 4.** Siamese Network with FaceNet as a feature extractor.

A pre-trained FaceNet model [35] was used as a feature extractor. Using a pre-trained model allowed us to utilize the transfer learning technique; transfer learning exploits the knowledge gained by a network from one task to complete another task. Network optimization requires collecting a considerable amount of data which increases the amount of training time and computational resources. Transfer learning reduces network training time dramatically by using a pre-trained network such as FaceNet to train the new model on a smaller dataset by exploiting its fine-tuned parameters and pre-trained weights.

The FaceNet model accepts images of size $160 \times 160$ and creates 128-dimensional vectors. The pre-processing applied to the Faces of subjects does not compromise the accuracy of FaceNet, because it was trained on the Labeled Faces in the Wild (LFW) benchmark dataset. After examining this dataset It was found that images contain faces and other features (e.g., hair, neck, a portion of the chest).

Each stream of the Siamese network is a FaceNet model that produces a 128-dimensional vector to represent an image. After that, we experimented with several distance metrics such as Hamming Distance, Euclidean Distance, Manhattan Distance (Taxicab or City Block), Minkowski Distance, Cosine Similarity to calculate the distance between the two images. After conducting several experiments, it was confirmed that Cosine Similarity and Euclidean Distance achieved the best performance.

Briefly, the Cosine similarity between two feature vectors $(u_i, v_i)$ is the cosine angle between two N-dimensional spaces [36]. It is calculated using the dot product of the two vectors divided by the product of the two vectors' lengths or magnitude, the exact formula is described in Equation (3).

$$similarity(u_i, v_i) = \frac{u.v}{||u|| * ||v||} = \frac{\sum_{i=1}^{N} u_i * v_i}{\sqrt{\sum_{i=1}^{N} u_i^2} * \sqrt{\sum_{i=1}^{N} v_i^2}} \tag{3}$$

Before calculating the Euclidean distance, it is often a good practice to normalize them, which means scaling values until the length/magnitude of vectors is one/unit length. L2 normalization was used to standardize the numerical values across all columns. The normalized vectors are passed to the distance metric; the Euclidean Distance. The

Equation (4) is used to illustrate how to obtain the Euclidean distance where $u_i$ and $v_i$ represent output vectors 1 and 2, respectively, $N$ is the size of the vectors.

$$d(u_i, v_i) = \left(\sum_{i=1}^{N} |(u_i - v_i)|^2\right)^{\frac{1}{2}} \tag{4}$$

After comparing the performance of both Euclidean distance and Cosine similarity, It can be concluded that both approaches almost maintain the same level of accuracy. However, Euclidean distance does not exhaust computational resources and consumes time like the Cosine similarity while maintaining accurate results. Therefore, the Euclidean distance was in our model. Finally, we augmented the model with a verification layer using a Sigmoid activation function. If the distance between the two feature vectors (pair of images under investigation) is equal to or greater than 1, the Sigmoid layer will round it to 1. If the distance between the two feature vectors is less than 1, the Sigmoid layer will round it to 0. The Sigmoid layer can transform the problem into a binary classification problem (2 classes). Either the pair of images are similar (class: 0) or not similar (class:1).

For the second modality, the remaining part of the input image mainly incorporates the body figure, and clothes were fed to the model [37]. The proposed model is a fusion between two state-of-the-art person re-identification models PCB and OSNet. Figure 5 visualizes the architecture of the model.

A strong backbone (OSNet) [16] network was used as a feature extractor . Features extracted from this network are particularly suitable for this modality because they combine heterogeneous and homogeneous features. Heterogeneous and homogeneous features are features extracted at multiple scales. Omni-scale features are a combination of heterogeneous and homogeneous features. Homogeneous features show that this image contains a young man, a white t-shirt, grey shoes. Those features are crucial, but they are not discriminative on their own. Heterogeneous features are more sophisticated, such as a specific logo on a t-shirt. Neither the white t-shirt nor the logo is sufficient on its own to identify a subject. Combining this information gives Omni-scale features. OSNet is formed by stacking the bottleneck layer-by-layer (multi-convolutional streams/residual block), using multiple convolutional streams (lite $3 \times 3$ convolutional networks with $1 \times 1$ convolutional network after and before the lite ones to reduce and restore feature dimensions), each detecting features at a certain scale. The resulting multi-scale feature maps are dynamically fused by weights generated by a novel unified aggression gate (a trainable mini-network sharing parameters across all streams), which can single out features from any particular scale or mix features from different scales, as necessary also to prevent over-fitting and to reduce the number of parameters and decrease computational cost, depth wise and pointwise convolutions were used in the lite $3 \times 3$ convolutional networks. OSNet is relatively a lightweight network and still achieving state-of-the-art performance. OSNet improves the order of magnitude compared to ResNet50. The trainable parameters in OSNet are 2.2 million, ResNet uses 24 million parameters.

The produced Omni-scale features are fed to PCB (Part-based Convolutional Baseline). PCB is widely used in the person re-identification task with different backbone networks. It extracts part-level features to provide fine-grained information. After the input image is fed to the backbone network (feature extractor) OSNet, the resulting 3D tensor **T** is divided into **p** slices (**p** = 6; the input image is divided into 6 parts so features will be extracted from 6 regions). After that, spatial average pooling is applied to the column vectors in each slice to produce **p c**_dimensional column vectors to be passed on to a convolutional layer that reduces the dimensionality. Finally, the column vectors are passed through a classification layer with different weights, each optimized by the softmax loss function to output the final decision of the model. The architecture of the model is illustrated in Figure 5. The output of the two modalities is combined as follows: now that we have for each query image the top 20 matches according to the first modality (soft bio-metrics region) and another 20 matches according to the second modality (body figure region). The output of both classifiers was

combined using score-level fusion. Each network is trained on its modality separately to output the top 20 similar images from the gallery for each query image and the similarity score based on the Euclidean distance between the query image and the predicted top 20 images. After that, gallery image matches from both sets are ranked (first modality matches and second modality matches). The top 20 matches are produced by comparing the 40 images (output of both sets) based on the Euclidean distance.

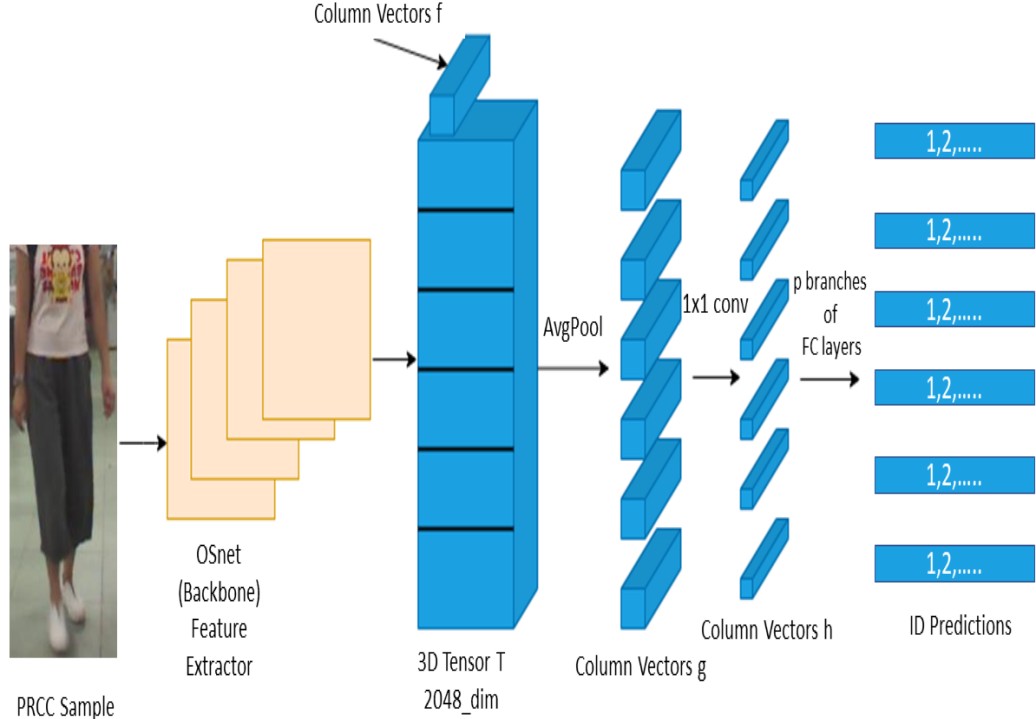

**Figure 5.** Illustration of the proposed model. OSNet acts as a feature extractor for the PCB, OSNet outputs a 3D tensor T. T is then partitioned into 6 parts each part consists of several column vectors f. Then an average pooling layer is applied to f to average each part into a column vector g. Then the spatial dimensions of g are reduced using $1 \times 1$ convolutional layer producing column vectors h which are fed into fully connected layers to predict the ID of the subject.

## 4. Experimental Results and Analysis

### 4.1. Experiment Setup

This section describes the experiments conducted. Google Colaboratory [38], often referred to as Google Colab was used for training models. Google Colab is a product from Google Research. A Jupyter Notebook with computing resources, including GPUs, is available on Google Colab. It contains a wide range of GPUs, such as Nvidia K80s, T4s, P4s, and P100s. There is no way for a user to choose the GPU, computing, and memory resources. Colab allocates resources to users according to a scheduling scheme that prioritizes interactive users over long-running computations causing resource fluctuation.

Part of our implementation was based on Torchreid [39], Torchreid is a software library built over Pytorch [40]. It allows fast development, end-to-end training as well as evaluation of deep re-identification models. Torchreid is a framework that includes unified data loaders for 15 of the most popular and widely used datasets in the field of person re-identification (video and image domains are available). Moreover, it provides streamlined pipelines and implementation for state-of-the-art deep learning models as well as recent models. In addition, it allows the usage of pre-trained models to facilitate reproducing results in future research.

*4.2. Evaluation Protocol*

The evaluation metrics used to evaluate the performance of each algorithm are:

- Cumulated Matching Characteristics (CMC): shows the probability that a query identity matches the candidate lists retrieved by the model, thus this evaluation metric is valid only for supervised learning algorithms since it relies on comparing the model's prediction with the ground truth [21]. As an example, consider a simple single-gallery-shot setting in which each gallery identity is only present once. Using the ground truth of the dataset the algorithm will rank all the gallery samples based on their distances compared to the query from small to large, and the CMC top-k accuracy is which is a shifted step function as shown in Equation (5). The final CMC curve is computed by averaging the shifted step functions over all the queries.

$$Acc_k = \begin{cases} 1 & \text{if top-k ranked gallery samples contain the query identity} \\ 0 & \text{otherwise} \end{cases} \tag{5}$$

- Mean Average Precision (mAp): evaluates the overall performance of the model. The average precision calculates the area under the precision-recall curve for each query, mAp calculates the mean of all queries. CMC curve combined with mAp can give a clear representation of the model's performance. Figure 6 shows the top five ranks of two different models on the same query image. Both models were able to retrieve two correct matches in the first five ranks, but model **b** was able to achieve them earlier. The mean average precision shows this difference. Table 2 shows the calculation of precision on ranks 1 through 5 for modela **a** and **b**, respectively, using the precision Equation (6) as well as the mean average precision using Equation (7)

**Table 2.** Precision on ranks 1 to 5 and mAP for models **a** and **b**, respectively.

| Model | Rank-1 | Rank-2 | Rank-3 | Rank-4 | Rank-5 | mAP |
|-------|--------|--------|--------|--------|--------|-----|
| Model A | 1/1 | 1/2 | 1/3 | 1/4 | 1/5 | 0.496 |
| Model B | 1/1 | 2/2 | 2/3 | 2/4 | 2/5 | 0.712 |

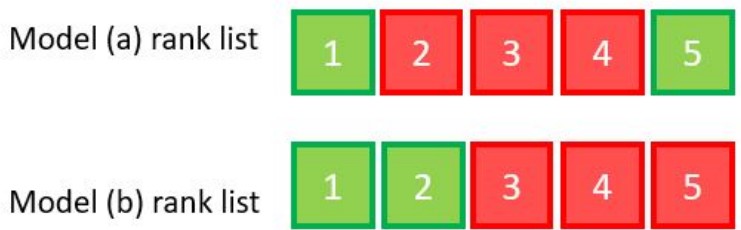

**Figure 6.** Example to illustrate the importance of mAp. Two models are represented (Model **a**, Model **b**). True matches of each model are represented in green boxes and false matches in red boxes.

$$Precision = \frac{TP}{TP + FP}; \tag{6}$$

*TP*: true positives, *FP*: false positives

$$mAP = \frac{\sum_{q=1}^{Q} P(q)}{Q} \tag{7}$$

where *Q* is the number of ranks, *P(q)* is the precision of each rank.

### 4.3. Experiments

A series of experiments to evaluate our model and compare it to some state-of-the-art models are presented. The experiments conducted used the following models:

- HACNN.
- MLFN.
- SCRNet.
- MuDeep
- ResNet50.
- PCB with a backbone network ResNet50: The Part-based Convolutional Baseline (PCB) network was utilized to conduct this experiment. The key idea behind PCB is to partition the produced column vector of an image to **p** equal partitions. Empirically PCB was always used with **p** = 6. As mentioned the idealized human figure approximately eight heads tall. The first modality represents two heads which makes the remaining column vector contain around six equal partitions. The presented model utilizes **p** = 6. As mentioned earlier PCB network needs the backbone to act as a feature extractor. This experiment employs ResNet50 because of its outstanding performance and its relatively compressed architecture. The ResNet50 model used was the pre-trained one which exploits the pre-trained parameters on ImageNet.
- PCB with ResNet50 as a backbone network augmented with RPP module: As mentioned one of the main contributions represented in [18] is augmenting the PCB network with a refined part-pooling (RPP) module. The RPP module is an attempt to relocate outliers. Outliers are part of the feature mAp inevitably located in the wrong partitions. RPP will measure the similarity between parts to assign each part to its correct partition, as shown in Figure 7.

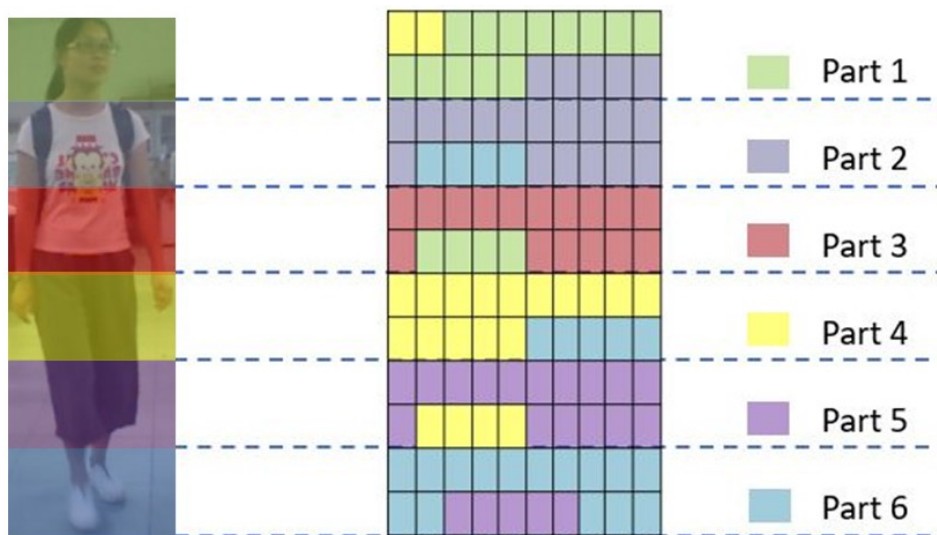

**Figure 7.** Visualization of outliers within parts. On the left tensor **T** is illustrated equally partitioned to 6 horizontal stripes *p* = 6. On the right, every column in **T** is shaded with the color of the closest part. Outliers can be seen in part 1 as the yellow and purple parts that do not match the color of part 1 (green).

- OSNet.
- PCB with OSNet as a backbone network: Since OSNet proved to be a strong feature extractor in the person re-identification task. In this experiment, we proposed augmenting the PCB network with OSNet as the feature extractor to improve the performance. OSNet is a lightweight network compared to ResNet50. In addition to improving the model's overall performance, the training time is also reduced.
- The proposed architecture utilized two separate neural networks for each modality. For the first modality (faces), the FaceNet Siamese model was trained on the extracted

facial features from the PRCC dataset using the Faceboxes detector. For the second modality (body), the second model that fuses OSNet and PCB was trained on the PRCC dataset after removing the soft bio-metrics region extracted for the first modality.

### 4.4. Validation of Proposed Model

To achieve a fair comparison between different methods, the setup of all experiments was standardized. All models trained for 40 epochs, with a learning rate of 0.0003 and an Adam optimizer. Models trained on the PRCC dataset. Models are fed the entire input image except for our proposed model. Before feeding the image to our model, the image is pre-processed to extract both modalities (face and body).

Figure 8 demonstrates some of the results. The first row of images shows a query image and the model's prediction (retrieved images), our model was able to perform very well in the case of moderate clothes changes. The second sample (second row) shows that the model can perform well under pose variation. The third sample (third row) examines our model under different conditions: pose variation, illumination changes, and moderate cloth variation.

Figure 9 shows another sample of the results. The two samples show that our model's performance might degrade in some special cases. For example, in Figure 9 sample 1 (first row) matched a different person just because of clothes color. However, it is clear that still there were positive matches which will increase if more than 10 ranks were retrieved. Using the first 20 ranks of images in forensic search still represents a huge improvement in efficiency.

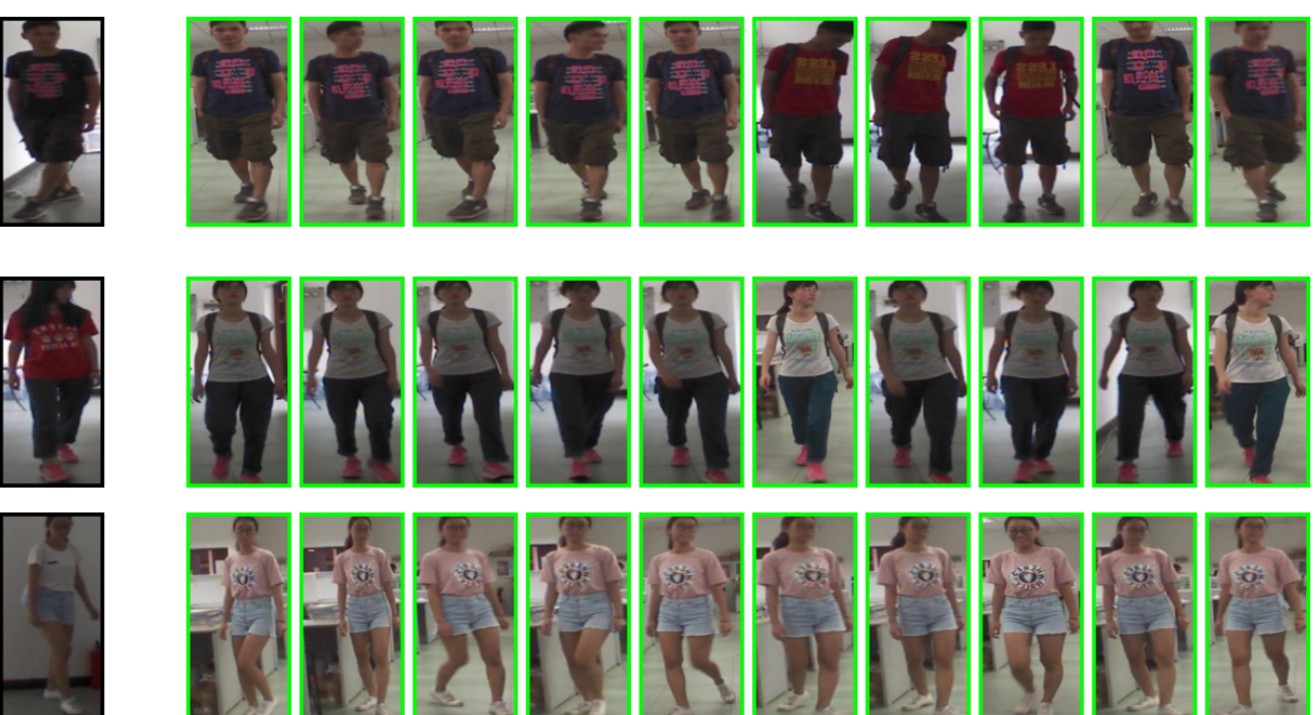

**Figure 8.** Sample 1 of results: 3 query images and our model's predicted matches from the gallery (Green frames indicate correct matches).

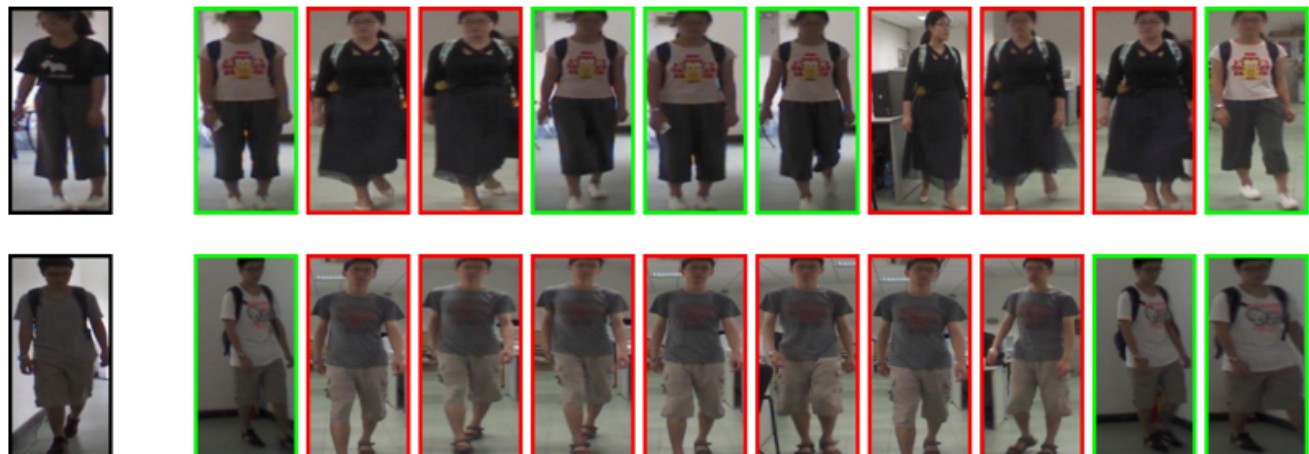

**Figure 9.** Sample 2 of results: 2 query images and our model's predicted matches from the gallery (Green frames indicate correct matches, Red frames indicate incorrect matches).

### 4.5. Results

Results are illustrated in the Table 3. After conducting a considerable number of experiments, we can confidently state that state-of-the-art models such as MuDeep and ResNet50 performance will dramatically deteriorate when trained on datasets with clothes variation. Other observations worth mentioning:

- ResNet-50 model performed relatively better than MuDeep (from Table 3 MuDeep Rank-1 accuracy: 58.5%, ResNet50 Rank-1 accuracy: 62.9%) which is why it was used as a backbone network to the PCB achieving 59.2%.
- To improve the performance of ResNet50 with PCB, the model was augmented with the RPP module, but it seemed to harm the model's performance decreasing the accuracy from 59.2 % to 51.2% accuracy on Rank-1.
- The outstanding performance of OSNet producing an accuracy of 63.6% on Rank-1 was the inspiration to replace the ResNet-50 used as a feature extraction for PCB with OSNet. This combination improved the accuracy on Rank-1 to 73.5%.
- The proposed architecture produced an accuracy of 81.4% on Rank-1. Our model was able to outperform some of the recent state-of-the-art approaches such as HACNN, MLFN and SCRNet in terms of accuracy and mAP.
- Our model does not consume an enormous amount of training time due to its lightweight architecture and exploiting pre-trained models.

**Table 3.** Performance (%) of our approach compared to other methods on the PRCC dataset.

| Method | Rank-1 | Rank-5 | Rank-10 | Rank-20 | mAP | Training Time |
|---|---|---|---|---|---|---|
| HACNN | 64.1 | 68.1 | 70.2 | 72.1 | 41.5 | 2:48:49 |
| MLFN | 50.2 | 56.1 | 59.4 | 62.7 | 29.6 | 2:54:21 |
| MuDeep | 58.5 | 64.6 | 67.6 | 71.6 | 34.1 | 3:46:54 |
| SCRNet | 78.1 | 79.5 | 81.7 | 83.2 | 58.2 | 1:57:32 |
| ResNet50 | 62.9 | 67.1 | 68.3 | 70.7 | 35.5 | 1:49:11 |
| PCB + ResNet50 | 59.2 | 66.4 | 69.7 | 71.4 | 31.8 | 2:14:49 |
| PCB + ResNet50 (RPP) | 51.2 | 52.3 | 52.5 | 53.4 | 28.0 | 2:47:12 |
| OSNet | 63.6 | 67.6 | 70.4 | 73.2 | 33.7 | 1:12:08 |
| PCB + OSNet | 73.5 | 75.1 | 75.8 | 76.8 | 48.6 | 2:10:01 |
| Our model | 81.4 | 82.3 | 83.1 | 83.7 | 60.2 | 2:35:42 |

## 5. Conclusions

In this paper, we propose an approach to address the long-term cloth changing person re-identification problem. We started by choosing a suitable dataset (PRCC dataset) to meet this challenge. After that, we constructed a model based on a multimodal approach. Initially, faces was the first modality; however, faces by themselves did not produce enough features. Therefore, the facial features were extended to include hair, neck, shoulder, a portion of the chest. Since the previously mentioned features do not tend to change hugely at different times, we combined them. The second modality was the remaining body figure. We constructed a unique model for each modality: a two-stream Siamese network with FaceNet as a feature extractor and a PCB network with OSNet as a feature extractor. The output of both networks was fused to produce the final decision of the model. Compared to several state-of-the-art approaches, there is a significant increase in performance. When examining the results of our model, it was able to match people with various kinds of clothes, which was difficult with the previous models. The experiments also show that the facial features were more accurate than the body features when there is extreme change of clothes. The conducted experiments helped identify some limitations; our model is prone to over-fitting due to the small size of the dataset. The model's performance would improve with a large-scale data set.

## 6. Future Work

The future projection is to experiment with the sketch contour modality provided by the PRCC dataset. It will help represent the body figure of a subject regardless of their clothes. Another projection is to investigate different loss functions that can be beneficial to our model, for example, Triplet loss [15]. Triplet loss is a loss function widely used in machine learning algorithms. Previous research suggests that Triplet loss improves the performance; it compares a baseline input to both a positive and negative pair. As a result, the model minimizes the distance between positive (similar image pairs) and maximizes the difference between negative (different image pairs) [41].

**Author Contributions:** Conceptualization, M.A.-M.S.; methodology, N.S.; validation, M.A.-M.S.; investigation, N.S.; writing—original draft preparation, N.S.; writing—review and editing, M.A.-M.S. and M.A.A.E.G.; supervision, M.A.-M.S.; funding acquisition, M.A.A.E.G. All authors have read and agreed to the published version of the manuscript.

**Funding:** This research was funded by TU Darmstadt.

**Institutional Review Board Statement:** Not applicable.

**Informed Consent Statement:** Not applicable.

**Data Availability Statement:** PRCC Dataset [26] is a publicly available dataset: https://www.researchgate.net/publication/345351618_Learning_Shape_Representations_for_Person_Re-Identification_under_Clothing_Change, accessed on 1 January 2022.

**Conflicts of Interest:** The authors declare no conflict of interest.

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
