# Peer review of "Multi-Modal Long-Term Person Re-Identification Using Physical Soft Bio-Metrics and Body Figure"

_applsci, doi:10.3390/app12062835_

Round 1

Reviewer 1 Report

The paper describes a Siamese network for person re-identification specifically for long term when clothing changes over time. The paper is well written in general, and the figures are good. The results look promising and there is good benchmarking against other methods. There are a number of areas that can be improved these are as follows:

There is discussion regarding distance metrics and testing (around line 300), this is described but would be better if there were metrics and some results to show you results in terms of accuracy and computational resource.

It would also be interesting to know which features provided the most use for re-id, does using facial features in part of the network increase this over the other methods which take the entire body as the input.

Formula 3 and 4, need an explanation for the parameters in each equation.

Section 3.2 the experiments there is a lot of writing about other methods, this would be better placed in the related work so that this section is primarily about the experiments.

Are the section number correct as they start at 0?

On line 499 it says your model had a 76.4% accuracy on Rank-1, I can see this value in the results shown in table 2, is this different metric as it is not clear and require clarification.

Author Response

We appreciate your precious time to review our paper and provide valuable insights and comments to enhance our manuscript. It was your valuable and insightful comments that led to possible improvements in the current version. The authors have carefully considered the comments and tried our best to address every one of them. You can find a detailed reply on each comment attached below.  "Please see the attachment."

Best regards, 

Reviewer 2 Report

Dear authors,

congratulations on your effort. Even though I disagree with some of the opening paragraphs being less academic than usual for academic papers, the work is solid.

Some individual comments:
- Unlike [3] Omni-scale features include heterogeneous  -> Unlike Xuelin Qian et.al. [3], Omni-scale features...
- Lines 115-116 should be merged
-   Yifan Sun et.al [11] the work proposed in this paper depends on part-level features. ???
- PCB has to be defined in its first appearance
- Line 365 there is a random '
- All figures and tables should be referenced within text before they appear.
- 4.1 Conclusions

Please fix all similar issues within the paper to increase its readability and academic stature. Other than that I like the idea and execution.

Author Response

(The authors gave the same response as above.)

Reviewer 3 Report

The authors present the article entitled “Multi-modal Long-term Person Re-identification Using Physical Soft Bio-metrics And Body Figure.” The article is hard to read and needs a hard revision according to the following concerns:

Avoid using first-person sentences. Use third-person sentences or passive voice instead.

The abstract section must be carefully checked. Some sentences are written starting with the exact words (i.e., “In our work”). Please rewrite it in a technical language for better comprehension.

Please correct the numbering of the section names. I argue to the authors to read carefully the guide for authors provided by the journal.

Lines 24-27: These two sentences can be merged. They are defining Person re-identification topic.

“re-id” is not defined before.

Lines 22-24: Is this sentence the objective of the manuscript? I suggest mentioning the objective manuscript next to the contributions in the Introduction section. The objective must be clear, and it has to highlight the novelty of the work.

Related work section should be entirely restructured instead of mentioning directly what has been done by the authors in the literature. A detailed study is required to find out the motive and novelty of the present work by extensively comparing the presented literature in the manuscript.

In the introduction, line 40, it can be included discriminative features extraction by considering the next references: A high-accuracy mathematical morphology and multilayer perceptron-based approach for melanoma detection; A novel method for measuring subtle alterations in pupil size in children with congenital strabismus; Efficient single image dehazing by modifying the dark channel prior; A new approach for motor imagery classification based on sorted blind source separation, continuous wavelet transform, and convolutional neural network.

Figure 1 is already presented before in the work of [9]. What is the justification for presenting exactly the same figure in this work?

Line 285: This must be presented in the Related works section.

Sub-section 3.1: Please add the mathematical and explain in detail the evaluation of the proposed method.

The experiments and results section is hard to follow and is extensive. It can be presented in two sections: Results and discussion for better comprehension. Please check the instructions for authors. In the discussion section, it is recommended to include a table that compares the findings of the work vs the already reported in the art stat.

Change the name to a conclusion for the last section and do not use sub-section headers.  

Some subsections are named as the section, but separately, i.e. Conclusions and Future Work. Please restructure the sections and subsections' names.

Avoid the references of proceedings, conferences, and journals that are outside of the JCR index since, for example, conferences are not well per-reviewed.

Balance the references between journals, there are many of this topic out of IEEE, for example, Elsevier and Wiley. Use Scopus or WOS to attempt this task.

Author Response

(The authors gave the same response as above.)

Round 2

Reviewer 1 Report

Thankyou for making the substantial changes to the manuscript, it is much improved in line with the reviewers comments.

Reviewer 3 Report

The manuscript has been greatly improved, it can be accepted for publication